# An Effective Strategy for Template-Free Electrodeposition of Aluminum Nanowires with Highly Controllable Irregular Morphologies

**DOI:** 10.3390/nano12091390

**Published:** 2022-04-19

**Authors:** Heng Wang, Guo-Min Li, Bing Li, Jing-Lin You

**Affiliations:** 1School of Mechanical and Power Engineering, East China University of Science and Technology, No.130 Mei Long Road, Shanghai 200237, China; wh1wh2wh3wh@163.com (H.W.); liguominyumifei@163.com (G.-M.L.); 2State Key Laboratory of Advanced Special Steel, School of Materials Science and Engineering, Shanghai University, Shanghai 200444, China

**Keywords:** Al nanowires, template-free electrodeposition, ionic liquid, irregular morphologies

## Abstract

Aluminum nanowires with irregular morphologies were prepared by template-free electrodeposition from a room-temperature chloroaluminate ionic liquid. The effects of the diffusion condition and deposition potential on the morphologies of Al nanowires were investigated. The decrease of diffusion flux leads to the formation of particular segmented morphologies of Al nanowires. A dynamic equilibrium between the electrochemical reaction and the diffusion of Al_2_Cl_7_^−^ results in the current fluctuation and the periodical variation of diameters in the Al nanowires growth period. Al nanowires with several kinds of morphologies can be controllably electrodeposited under a restricted diffusion condition, without using a template. Increasing the overpotential shows the similar influence on the morphology of Al nanowires as the decrease in diffusion flux under the restricted diffusion condition. Most of the segmented Al nanowires have a single crystalline structure and grow in the [100] orientation. This work also provides a new strategy for the fabrication of nanowires with highly controllable irregular morphologies.

## 1. Introduction

One-dimensional (1D) nanostructures have tremendous potential applications in various fields because of their distinct chemical and physical properties [1]. In recent years, a series of nanowires with irregular morphologies (for example, Jagged-like [2,3], screw-like [4,5] and bead-like [6] nanowires) have exhibited exceptional properties in photocatalysis [7], electrocatalysis [8,9,10], photoelectrocatalysis [11], lithium-ion batteries [12,13], sensors [14] and field emission [15]. The uneven surfaces and high-index facets of the nanowires have been demonstrated to be the key of their unprecedented performance [16]. Therefore, precise design and control of the size and morphology during the growth of nanowires play critical roles in imparting new and enhanced functionality.

To date, several kinds of methods, such as chemical vapor deposition [17], thermal evaporation [18,19], microplasma-based synthesis [20], electrospinning [21,22], carbothermal reduction method [23] and microcontact printing [24] have been attempted to fabricate semiconductor and metal oxide nanowires with irregular morphologies. For metal nanowires, asymmetrical etching of screw-like silver nanowires [25] and a class of jagged Pt-based nanowires prepared by wet chemical methods [26,27,28,29,30,31] have been reported. A simple and controllable route for fabricating various metal nanowires with irregular morphologies is still one of the crucial challenges in materials research.

As a typical bottom-up process, template-assisted electrodeposition is a versatile method for synthesis of metal and semiconductor nanowires [32,33]. Nanowires electrodeposited in template normally have a uniform diameter along their growing axis, since most nanochannels of common templates are straight and have nearly the same size. Besides plain nanowires, a few kinds of nanowires with periodically modulated diameters [34,35], multisegmented structures [36] and multilayered structures [37] have been fabricated in template-assisted electrodeposition by choosing special templates and altering deposition conditions. Nevertheless, the preparation of templates with irregular shaped nanopores seems rather difficult and costly [38].

Template-free electrodeposition is more advantageous than template-assisted electrodeposition, as the costly and time-consuming preparation and removal of the templates are not needed. Moreover, without the restriction of templates, it is possible to directly modulate the morphologies of the template-free electrodeposited nanowires by different approaches [39]. In the past decade, one-dimensional growth of Sn [40,41], Zn [42,43], Te [44,45], Co [46], Sb [47], Al [48,49] and their alloys [50,51,52,53,54,55] has been realized in template-free electrodeposition from specific ionic liquids. Recently, a pulse potential deposition technique has been successfully applied in template-free electrochemical fabrication of beaded aluminum sub-microwires [56], diameter-modulated Co-Zn/oxide wires [57] and screw-like PdPt nanowires [58]. The bead-like and screw-like morphologies are formed by periodically changing the applied overpotential, which is one of the critical factors in controlling the diameter of the template-free electrodeposited wires. However, in the above process of modulating the morphology of nanowires, a pulse power source is indispensable to precisely control the electrodeposition parameters.

In this work, aluminum nanowires with several kinds of morphologies were prepared by template-free electrodeposition from a room-temperature ionic liquid. The morphologies of aluminum nanowires can be simply tuned through modifying diffusion conditions by using different sizes of insulation rings on the working electrode. Varying the deposition potential also influences the morphology of Al nanowires in the restricted diffusion condition. This work also provides a new strategy for the template-free electrodeposition of nanowires with highly controllable irregular morphologies.

## 2. Materials and Methods

### 2.1. Materials and Synthesis of Al Nanowires

All the experiments were carried out inside an argon-filled glove box with O_2_ and H_2_O content lower than 1 ppm. The aluminum chloride (AlCl_3_)–1-butyl-1-methylpyrrolidinium chloride (BMPC) ionic liquid was prepared by carefully adding AlCl_3_ (Alfa Aesar, Shanghai, China, 99.9%) powder into BMPC (Sigma-Aldrich, Darmstadt, Germany, 99%) in a glass beaker with the mole ratio of AlCl_3_:BMPC of 2:1 and then continuously stirring for 24 h. A conventional three-electrode cell, as shown in Figure 1, was used for the electrochemical experiments. The reference electrode (RE) was prepared by immersing a 0.5 mm diameter Al wire (Alfa Aesar, Shanghai, China, 99.9995%) in the AlCl_3_- BMPC ionic liquid, which was contained in a fritted glass tube. The counter electrode (CE) was a 0.1 mm–thick Al plate (Aladdin, Shanghai, China, 99.99%). For the electrodeposition under unrestricted diffusion condition, a mechanically polished 0.1 mm–thick Cu plate (99.9%) was cut into a square with a 0.8 ± 0.2 mm–wide “tail” on top and served as the working electrode (WE). For the electrodeposition under restricted diffusion condition, the square Cu plate was used as the working electrode, with one side covered by a polytetrafluoroethylene (PTFE) insulation ring and the other side covered by a PTFE insulation plate, as shown in Figure 1. The Cu plate, PTFE insulation ring and PTFE insulation plate were fixed together by using a double-sided adhesive tape (AS ONE, Shanghai, China, No. 5000NS) with a circular Cu surface exposed. When the insulation ring was thicker than 2 mm, stainless-steel clamps were also used to fix the three parts instead of the double-sided adhesive tape and could reach the same results. When the thickness of insulation ring was 0.1 mm, a 0.1 mm–thick double-sided adhesive tape with a circular hole covered the square Cu plate and served as the insulation ring directly. The lateral diffusion of ions was restricted by the insulation ring during the electrodeposition processes. Before experiment, all the electrodes were degreased and cleaned with ethyl alcohol in an ultrasonic cleaner. The Cu plate was further treated by dipping in dilute sulfuric acid for 5 min to get rid of the oxide on the surface, and then washing with deionized water and drying. The electrochemical experiments were performed with a PARSTAT2273 (AMETEK, Berwyn, PA, USA) potentiostat/galvanostat controlled by PowerSuite software package (AMETEK, Berwyn, PA, USA, Version 2005). After electrodeposition, the insulation ring was removed from the electrode as soon as possible, and the Cu plate with deposits was rinsed in acetonitrile for 5 times.

The influences of the thickness of insulation ring, the inner diameter of insulation ring and the overpotential on the morphology, as well as the growth process of Al nanowires electrodeposited in the restricted diffusion condition, were studied.

### 2.2. Characterization Techniques

The template-free electrodeposited Al nanowires were characterized by scanning electron microscopy (SEM, FEI, Hillsboro, OR, USA, NOVA NanoSEM 450 and ZEISS, Oberkochen, Germany, GeminiSEM 500), transmission electron microscopy/high-resolution transmission electron microscopy (TEM/HRTEM, JEOL, Akishima, Japan, JEM-2100 and JEM-2100F) and selected area electron diffraction (SAED, JEOL, Akishima, Japan, JEM-2100 and JEM-2100F).

## 3. Results and Discussion

### 3.1. Electrodeposition in the Unrestricted Diffusion Condition

In our previous paper [49], we reported that single-crystalline Al nanowires can be template-free electrodeposited from the AlCl_3_–BMPC room-temperature ionic liquid and proposed the electrochemical 2D nanofilm–3D nanonuclei–1D nanowires (EC 2D–3D–1D) growth mechanism. Al_2_Cl_7_^−^ is considered as the reducible species in chloroaluminate ionic liquids, and the electrochemical reaction is carried out according to Equation (1) [59].
(1)$$4Al_{2}Cl_{7}{}^{-}+3e^{-}\to \mathrm{Al}+7\mathrm{AlC}l_{4}{}^{-}$$

In the electrodeposition experiments using various sizes of Cu plates in the unrestricted diffusion condition, some interesting phenomena in the chronoamperometry curves and morphologies of the Al nanowires appeared and attracted our attention. The chronoamperometry curves recorded during the electrodeposition showed an obvious difference as the side length of the square Cu plate altered from 3 to 7 mm. The current becomes fluctuant in the growth stage of Al nanowires when the size of Cu plate is larger than 5 × 5 mm, as shown in Figure 2a. In the SEM observation, the electrodeposited Al nanowires on the marginal part of the 7 × 7 mm–sized Cu plate show relatively smooth surfaces, while the Al nanowires on the central part exhibit the particular segmented morphology, as presented in Figure 2b,c. In fact, the two parts of the Cu electrode were immersed in the same ionic liquid and applied the same potential. The only perceptible distinction between the two parts is that the marginal part can easily acquire more active ions (Al_2_Cl_7_^−^) from the lateral diffusion. Therefore, the diffusion may have a critical influence on the morphology of deposits.

### 3.2. Electrodeposition in the Restricted Diffusion Condition

To investigate the nature of the unusual phenomena, we designed and performed a serial of experiments for electrodeposition of Al nanowires under restricted diffusion condition. As depicted in Figure 3, the lateral diffusion is retarded by attaching an insulation ring to the Cu plate in the restricted diffusion condition. The chronoamperometry curves recorded on Cu plate with different thicknesses of insulation ring during electrodeposition at −1.2 V (vs. Al^3+^/Al)/25 °C are presented in Figure 4. The shape of chronoamperometry curve recorded on Cu plate with a 0.1 mm–thick insulation ring is very similar to those recorded in the unrestricted diffusion condition. Once the potential is supplied, the current increases sharply to a peak value corresponding to the 2D growth of Al nanofilm. Then the current decreases gradually to a trough, corresponding to the formation and gradually thickening of the Al_2_Cl_7_^−^ depleted zone [49]. Since the lateral diffusion is retarded by the insulation ring, the Al_2_Cl_7_^−^ depleted zone may expand more quickly as the thickness of insulation ring increases; hence, the currents decrease more sharply in the curves of Figure 4b. Then the current reaches a stable plateau corresponding to the growth of Al nanowires. The apparent current fluctuation starts before the trough and continues to the initial stage of the Al nanowires growth period. The fluctuation should be related to the unsynchronization in nucleation and growth of Al nanowires, similar to the unrestricted diffusion condition. As the thickness of insulation ring increases to 0.5 mm, the current fluctuates all through the Al nanowires growth period. When the insulation ring thicker than 0.5 mm is used, the corresponding current also fluctuates similarly as shown in Figure 4b. It can be proved that the decrease of the lateral diffusion leads to the fluctuation of current during the growth of Al nanowires. Since the diffusion is retarded by the insulation ring, the current density during the Al nanowires growth period under restricted diffusion condition is much smaller than the one under unrestricted diffusion condition at the same potential, and the current density gradually decreases as the thickness of the insulation ring increases.

Since the diffusion flux at the electrode surface is proportional to the current, the current density ratio of the plateau in the restricted diffusion condition to that in the unrestricted diffusion condition can be used to quantitatively describe the decrease of diffusion caused by using the insulation ring. The average current density of the plateau recorded on the 5 × 5 mm–sized Cu plate in the unrestricted diffusion condition is −21.43 mA/cm^2^ according to the data in Figure 2a. As the thickness of insulation ring (inner diameter is 6 mm) increases from 0.1 to 3 mm in the restricted diffusion condition, the average current densities of the plateaus decrease from −11.84 to −5.06 mA/cm^2^ according to the data in Figure 4. Accordingly, the current density ratio of the plateau in the restricted diffusion condition to that in the unrestricted diffusion condition decreases from 55.25% to 23.61%, indicating that the diffusion flux of Al_2_Cl_7_^−^ decreases by 44.75% to 76.39% as the thickness of insulation ring increases from 0.1 to 3 mm. The average current densities of plateaus and the decrease of diffusion flux using different thicknesses of insulation ring are summarized in Table 1.

### 3.3. Effect of Insulation Ring Size

The morphologies of Al nanowires electrodeposited on the Cu plate with different thicknesses of insulation ring (inner diameter is 6 mm) at −1.2 V (vs. Al^3+^/Al)/25 °C are exhibited in Figure 5a–j. As the thickness of insulation ring gradually increases from 0.1 to 3 mm, the lateral diffusion decreases more and more sharply. As a result, the morphology of Al nanowires electrodeposited under the restricted diffusion condition changes accordingly. Since the lateral diffusion has more direct influence on the circumference than the central part of the circle, the marginal part (about 0.5 mm away from the circumference) and the central part (about 0.5 mm away from the center of the circle) also show perceptible different morphologies. Similar to the results in the unrestricted diffusion condition without using the insulation ring, most of Al nanowires on the marginal part of the Cu plate electrodeposited with a 0.1 mm–thick insulation ring show smooth surfaces. Meanwhile, the surfaces of Al nanowires on the central part of the same Cu plate become rough, as shown in Figure 5b. When the thickness of insulation ring rises to 0.5 mm, the diffusion amount of Al_2_Cl_7_^−^ further decreases, the Al nanowires electrodeposited on the central part of Cu plate exhibit the bead-chain-like morphology and the Al nanowires on the marginal part show rough surfaces, as presented in Figure 5c,d. A few of the spine-like Al nanowires appear on the central part of Cu plate, along with some bead-chain-like Al nanowires, as the 2 mm–thick insulation ring is used. As exhibited in Figure 5j, most of the Al nanowires on the central part of Cu plate display the spine-like morphology when the thickness of insulation ring is 3 mm. As the potential supplied in electrodeposition changes to −1.4 V (vs. Al^3+^/Al) and the thickness of insulation ring is 2 mm, the Al nanowires show the bead-chain-like and spine-like morphology on the marginal part, while there is a spine-like, with some fish-bone-like, morphology on the central part of Cu plate, as presented in Figure 5k,l.

As the inner diameter of the insulation ring is increased from 6 to 8 mm, the impact of the lateral diffusion on the central part of Cu plate further decreases; hence, the diffusion flux of Al_2_Cl_7_^−^ near the central part of Cu plate decreases accordingly. When a 0.1 mm–thick insulation ring with an 8 mm inner diameter is used, the chronoamperometry curve turns wavy in the Al nanowires growth period, as exhibited in Figure 6a, and the bellows-like Al nanowires can be obtained on the central part of Cu plate electrodeposited at −1.2 V (vs. Al^3+^/Al)/25 °C, as shown in Figure 6b. Al nanowires on the central part of the Cu plate exhibited the bead-chain-like morphology when a 0.5 mm–thick insulation ring with an 8 mm inner diameter was used, as presented in Figure 6c.

To sum up, the rule of the influence on the morphology of Al nanowires by diffusion can be defined as follows: the morphology turns from smooth surfaces to rough surfaces, and then it becomes bellows-like, bead-chain-like, spine-like and fish-bone-like as the diffusion flux decreases gradually. The periodical variation of Al nanowires’ diameters always coincides with the wavy current in the chronoamperometry during the growth of the Al nanowires, which should be related to the fluctuation of the diffusion amount of Al_2_Cl_7_^−^. The viscosity of AlCl_3_–BMPC (2:1 mole ratio) ionic liquid is 39.7 mPa s at 25 °C [49], which is much higher than the viscosity of ionic liquids commonly used for electrodeposition of Al (such as AlCl_3_–EMIC and AlCl_3_–BMIC) [60,61]. Moreover, the viscosity gradually increases with the decrease of the mole fraction of AlCl_3_ in this category of ionic liquids [60]. These features may result in a particular diffusion process near the interface in the electrodeposition of Al nanowires.

### 3.4. Effect of Overpotential

The chronoamperometry curves recorded on a Cu plate during the electrolysis at various potentials under the restricted diffusion condition with a 2 mm–thick insulation ring also show the fluctuation of current during the growth of Al nanowires, as presented in Figure 7. In the unrestricted diffusion condition, it was found that the diameter of the electrodeposited nanowires decreased as the deposition overpotential increased [48,49], while, in the restricted diffusion condition, it is observed that altering the supplied potential can also affect the morphology of Al nanowires. As exhibited in Figure 8, the bellows-like Al nanowires electrodeposited at −1.0 V (vs. Al^3+^/Al), the bead-chain-like Al nanowires electrodeposited at −1.1 V (vs. Al^3+^/Al), the spine-like Al nanowires (Figure 5l) electrodeposited at −1.4 V (vs. Al^3+^/Al) and the fish-bone-like Al nanowires electrodeposited at −1.6 V (vs. Al^3+^/Al) were acquired at 25 °C for 120 s. This indicates that the increase of the overpotential shows a similar influence on the morphology of Al nanowires as the decrease in diffusion flux under the restricted diffusion condition. As discussed above, reaching a dynamic equilibrium between the electrochemical reaction and the diffusion of Al_2_Cl_7_^−^ is critical in the growth process of the Al nanowires with segmented morphologies. Since the electrochemical reaction accelerates with the increase of overpotential, increasing the overpotential may leads to the similar equilibrium state as decreasing the diffusion flux of Al_2_Cl_7_^−^. After raising the temperature, we observed that the bamboo-like Al nanowires were found in the deposits at −1.2 V for 120 s at 40 °C, as presented in Figure 8d. Several Al nanowires with other special morphologies, such as spiral-like (as shown in Figure 8f,g) and sloping spine-like (as shown in Figure 8h,i), are also found in SEM observation.

### 3.5. Typical Morphologies

The typical morphologies of electrodeposited Al nanowires in SEM observation are exhibited in Figure 9. The nanowire with the segmented morphology can be considered as a chain of periodic repeated units. The shape of the repeated unit (between the two adjacent narrowest points) of the bellows-like, bead-chain-like, spine-like and fish-bone-like Al nanowires is flat hexagon, approximate regular hexagon, trapezoid and triangle, respectively. The length of the repeated unit (the space between the two adjacent narrowest points) is about 35 nm (bellows-like), 50 nm (bead-chain-like), 55 nm (spine-like) and 60 nm (fish-bone-like), respectively, as the electrodeposition temperature is 25 °C. When the electrodeposition is carried out at 40 °C, the length of the repeated unit can rise over 100 nm, since the electrochemical reaction accelerates at a higher temperature. The length of the repeated unit obviously increases with the decrease of diffusion or the increase of the overpotential. The diameter of the widest points (d_1_) and the diameter of the narrowest points (d_2_) of the fish-bone-like nanowire are marked in Figure 9. In most of the segmented Al nanowires, the diameter of the widest points (d_1_) is in the range of 50 to 120 nm. The increase of the electrodeposition temperature will lead to the augmentation of d_1_. As the morphology of the Al nanowires changes from the bellows-like to the fish-bone-like morphology, the ratio of d_1_/d_2_ increases from about 1.5 to around 2.4.

### 3.6. Growth Process

During the electrochemical reaction, Al_2_Cl_7_^−^ anions are consumed, and this is accompanied by the generation of AlCl_4_^−^ on the tips of the growing Al nanowires. In the unrestricted diffusion condition, when the diffusion of Al_2_Cl_7_^−^ is sufficient, the electrochemical reaction and the diffusion of anions can reach a delicate equilibrium in the stable growth period of Al nanowires. The concentration of Al_2_Cl_7_^−^, as well as the viscosity of the ionic liquid in the Al_2_Cl_7_^−^ depleted zone, stays in an appropriate range, and the Al_2_Cl_7_^−^ depleted zone steadily expands with the growth of the nanowires. Hence, the electrodeposited Al nanowires show the ordinary morphology with smooth surfaces. Meanwhile, in the restricted diffusion condition, when the diffusion amount of Al_2_Cl_7_^−^ is declined by using the insulation ring, the equilibrium between electrochemical reaction and the diffusion of anions is destructed. As the reaction progresses, the depleting of Al_2_Cl_7_^−^ anions and the accumulating of AlCl_4_^−^ anions near the nanowires and ionic liquid interface lead to the sharp increase of viscosity of the ionic liquid in front of the growing tips of the Al nanowires [56,60]. The diffusion coefficient of Al_2_Cl_7_^−^ varies with the large change of the local viscosity. Therefore, the diffusion of Al_2_Cl_7_^−^ is impeded near the interface, and the growth rate of the Al nanowires in the long-axis direction drops accordingly. Since there are some unreacted Al_2_Cl_7_^−^ anions between the adjacent tips of Al nanowires, the lateral growth of the tips can be maintained. Hence, the current density decreases, and the diameters of the growing tips increase in this period (period 1), as exhibited in Figure 9. In the next period (period 2), as the electrochemical reaction on the tips of Al nanowires becomes slow in period one, the amount of AlCl_4_^−^ produced by the reaction accordingly decreases. Meanwhile, the Al_2_Cl_7_^−^ anions continue diffusing near the interface. As a result, the viscosity of the ionic liquid in front of the growing nanowires’ tips consequently decreases, and then the diffusion of Al_2_Cl_7_^−^ accelerates near the interface, and the growth rate of the Al nanowires in the long-axis direction increases accordingly. Therefore, the current density rises and the diameters of the growing tips decrease in period two. The two periods are carried out alternately, similar to a negative feedback process, and lead to a dynamic equilibrium between the electrochemical reaction and the diffusion of Al_2_Cl_7_^−^. Consequently, the current fluctuates in the Al nanowires’ growth period, and the electrodeposited Al nanowires present the periodical varying morphologies under the restricted diffusion condition. The growth length of Al nanowires along the long-axis direction in period one (length one) and period two (length two) also affect the morphologies of Al nanowires. The length one and length two are approximately the same in the bellows-like and the bead-chain-like Al nanowires. Meanwhile, length two is about 2 times the length of length one in the spine-like Al nanowires and 4.5 times in fish-bone-like Al nanowires.

The formation of the Al nanowires under the restricted diffusion condition also accords with the three stages of the EC 2D–3D–1D growth process. As with the unrestricted diffusion condition, a shiny pre-deposited Al nano thin film can be observed on the Cu substrate in a short time after a negative potential is applied. As presented in Figure 10a, the Al film consists of uniform fine grains. Since the diffusion of Al_2_Cl_7_^−^ is reduced, the nucleation on the pre-deposited Al film starts after more than 10 s, when the potential is applied. After nucleation on the Al film, the Al nano-nuclei continue growing to form Al nanowires, as shown in Figure 10b. The Al nanowires on the marginal part of the Cu plate electrodeposited at −1.4 V/25 °C, with a 2 mm–thick insulation ring, can grow to lengths up to 40 μm in 300 s, with an average diameter of about 80 nm, reaching a high aspect ratio over 500. The average growth rate of Al nanowires at −1.4 V/25 °C is more than 0.13 μm/s and even higher at −1.6 V or at a higher temperature. The deposits show quite uniform morphology on a large scale, as presented in Figure 10d.

### 3.7. Growth Orientation

As determined in selected area diffraction patterns and high-resolution TEM images presented in Figure 11 and Appendix A, most of the segmented Al nanowires have a single crystalline structure and grow in the [100] orientation. Figure 8e shows that the tips of Al nanowires are pyramidal and expose the four equivalent planes of the {111} planes, which grow slower than the {100} planes. That is the typical face-centered cubic (FCC) crystal morphology of the [100] directional growth. On the ideal crystal morphology, the angle between the opposite planes on the pyramid is 70.53° and the angle between the opposite edges of the pyramid is 90°, which determines that the apparent angles on the tips of Al nanowires in SEM observation should be between 70.53° and 90°. The ideal cross-section of the [100] directional Al nanowires (namely the {100} planes) is square, suggesting that there are, at maximum, 1.41 times differences on the apparent diameters of Al nanowires, depending on the angle of view in SEM observation. The varying angle of view may also lead to distinguishing apparent morphology in SEM observation.

## 4. Conclusions

Aluminum nanowires with irregular morphologies can be directly electrodeposited from the AlCl_3_–BMPC room-temperature ionic liquid in the restricted diffusion condition by using an insulating ring. As the thickness of insulation ring with a 6 mm inner diameter increases from 0.1 to 3 mm, the diffusion flux of Al_2_Cl_7_^−^ decreases by 44.75% to 76.39%. The decrease of diffusion leads to the appearance of particular segmented morphologies. As the diffusion decreases gradually, the morphology of Al nanowires turns from smooth surfaces to rough surfaces and then becomes bellows-like, bead-chain-like, spine-like and fish-bone-like. Al nanowires with the morphology of rough surfaces, bead-chain-like and spine-like can be electrodeposited on the central part of Cu plate at −1.2 V/25 °C, using an insulation ring (inner diameter is 6 mm) with thicknesses of 0.1, 1 and 3 mm, respectively. A dynamic equilibrium between the electrochemical reaction and the diffusion of Al_2_Cl_7_^−^ results in the current fluctuation and the periodical variation of diameters in the Al nanowires growth period. An increase of the overpotential shows the similar influence on the morphology of Al nanowires as the decrease of diffusion in the restricted diffusion condition. The bellows-like, bead-chain-like, spine-like and fish-bone-like morphology of Al nanowire can be obtained on a Cu plate electrodeposited at −1.0, −1.1, −1.4 and −1.6 V, respectively, when a 2 mm–thick insulation ring with a 6 mm inner diameter is used. The formation of Al nanowires under the restricted diffusion condition also accords with the EC 2D–3D–1D growth process. Most of the Al nanowires with irregular morphologies are single crystalline structure and grow along the [100] orientation. Since the mechanism of this interesting growth behavior should be general in other ionic liquids with the similar properties, this work also provides a new strategy for fabrication of nanowires with highly controllable irregular morphologies.

## Figures and Tables

**Figure 1 nanomaterials-12-01390-f001:**
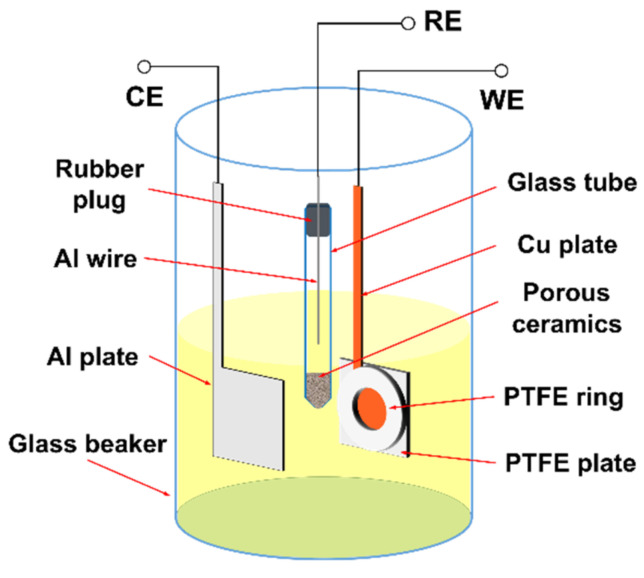
Schematic of the three-electrode cell used for the electrochemical experiments under the restricted diffusion condition.

**Figure 2 nanomaterials-12-01390-f002:**
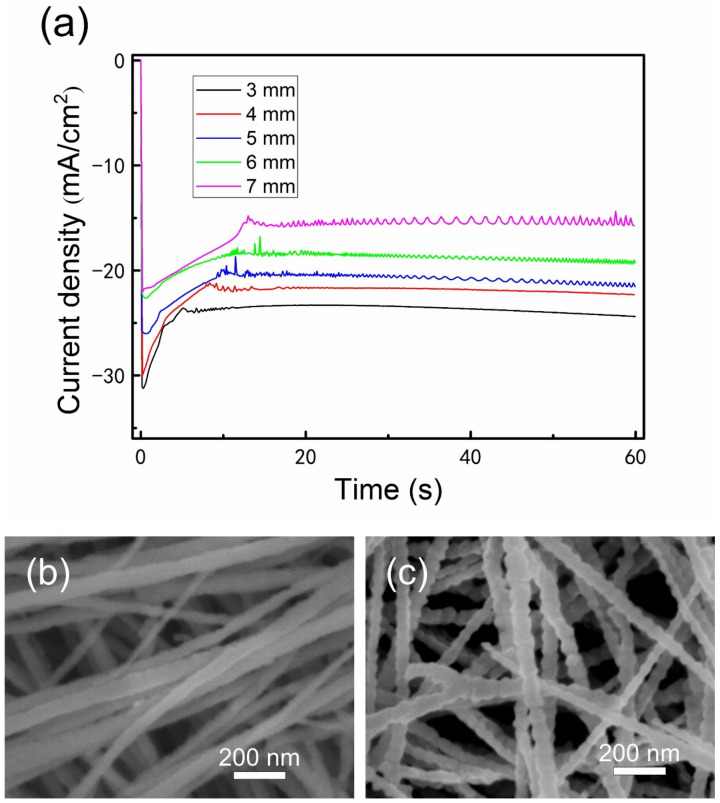
Potentiostatic electrodeposition of Al nanowires in AlCl_3_–BMPC (2:1 molar ratio) ionic liquid at −1.2 V/25 °C for 60 s under the unrestricted diffusion condition: (**a**) the chronoamperometry curves recorded on various sized Cu plates (the side lengths of squares are marked in the graph) during the electrodeposition, and SEM images of the deposits on the marginal part (**b**) and the central part (**c**) of the 7 × 7 mm–sized Cu plate.

**Figure 3 nanomaterials-12-01390-f003:**
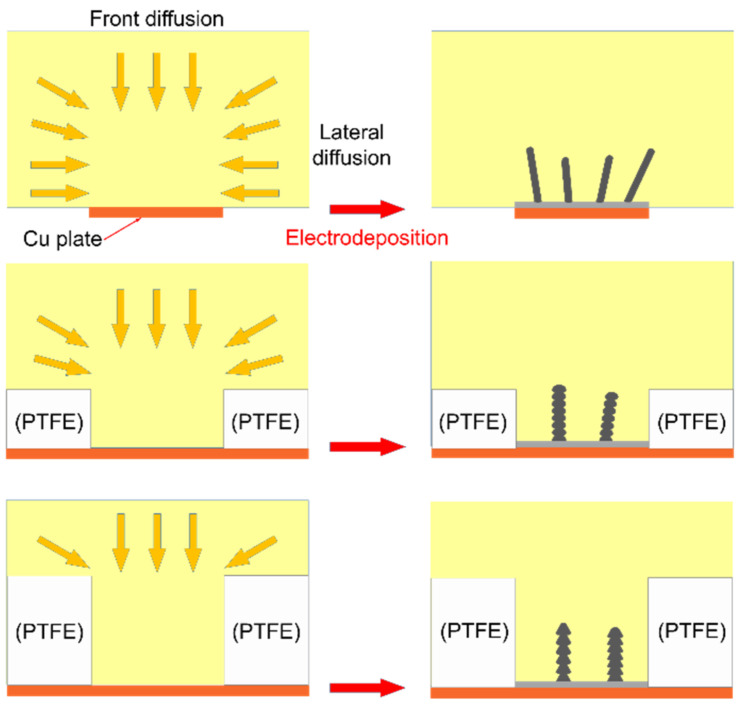
Schematic of the diffusion and morphology differences between the unrestricted diffusion condition and the restricted diffusion condition.

**Figure 4 nanomaterials-12-01390-f004:**
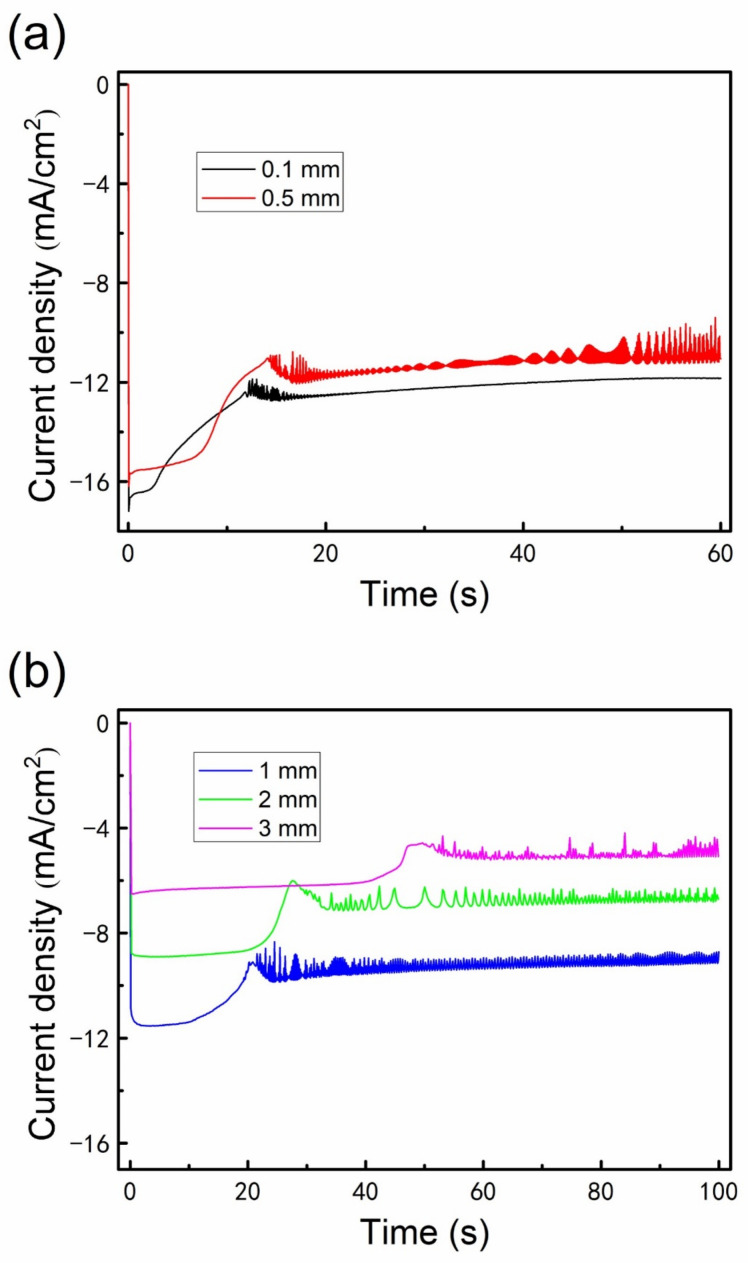
Chronoamperometry curves recorded on Cu plate with different thicknesses of insulation ring (inner diameter was 6 mm) during the electrodeposition of Al nanowires in AlCl_3_–BMPC (2:1 molar ratio) ionic liquid at −1.2 V/25 °C under the restricted diffusion condition: (**a**) 0.1 and 0.5 mm; (**b**) 1, 2 and 3 mm. Deposition time was (**a**) 60 s and (**b**) 100 s.

**Figure 5 nanomaterials-12-01390-f005:**
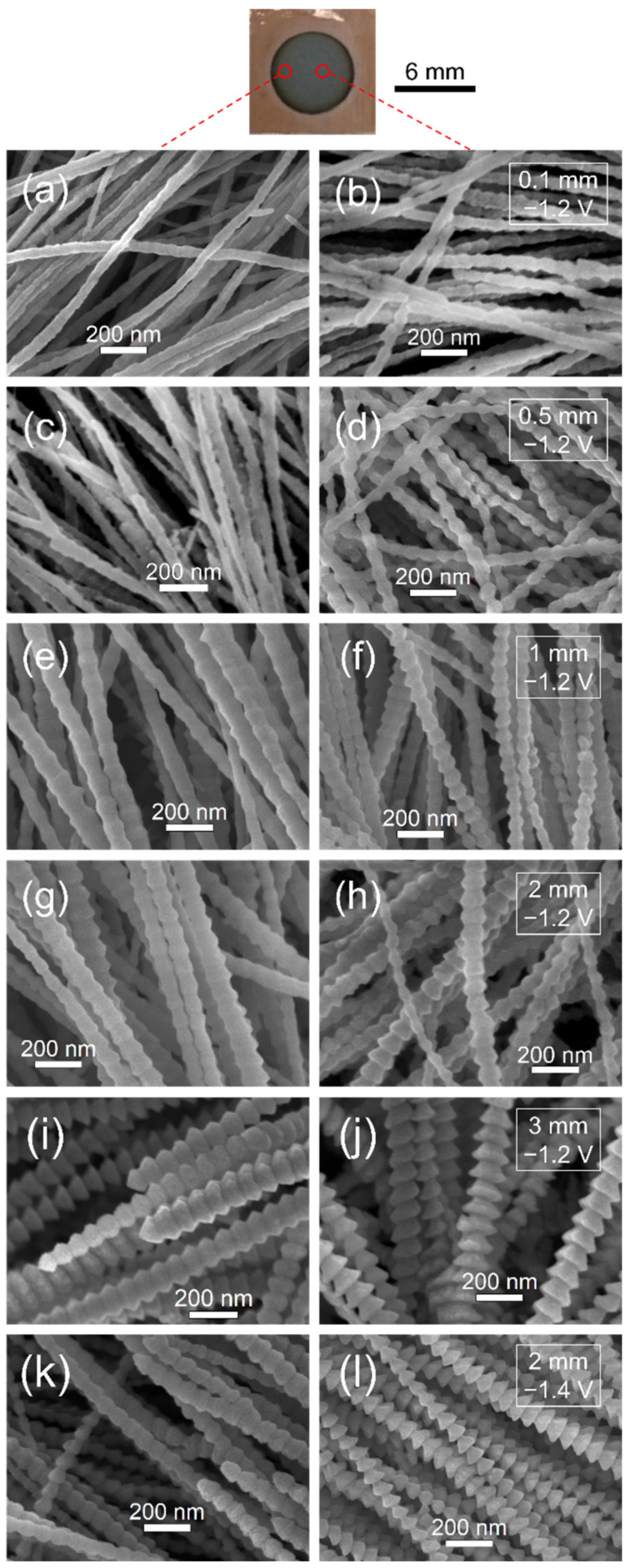
SEM images of Al nanowires on the marginal part (**a**,**c**,**e**,**g**,**i**,**k**) and the central part (**b**,**d**,**f**,**h**,**j**,**l**) of Cu plate electrodeposited with different thicknesses of insulation ring (inner diameter was 6 mm) at 25 °C: (**a**,**b**) 0.1 mm, (**c**,**d**) 0.5 mm, (**e**,**f**) 1 mm, (**g**,**h**,**k**,**l**) 2 mm and (**i**,**j**) 3 mm. Deposition potential was (**a**–**j**) −1.2 V and (**k**,**l**) −1.4 V. Deposition time was (**a**–**d**) 60 s and (**e**–**l**) 120 s.

**Figure 6 nanomaterials-12-01390-f006:**
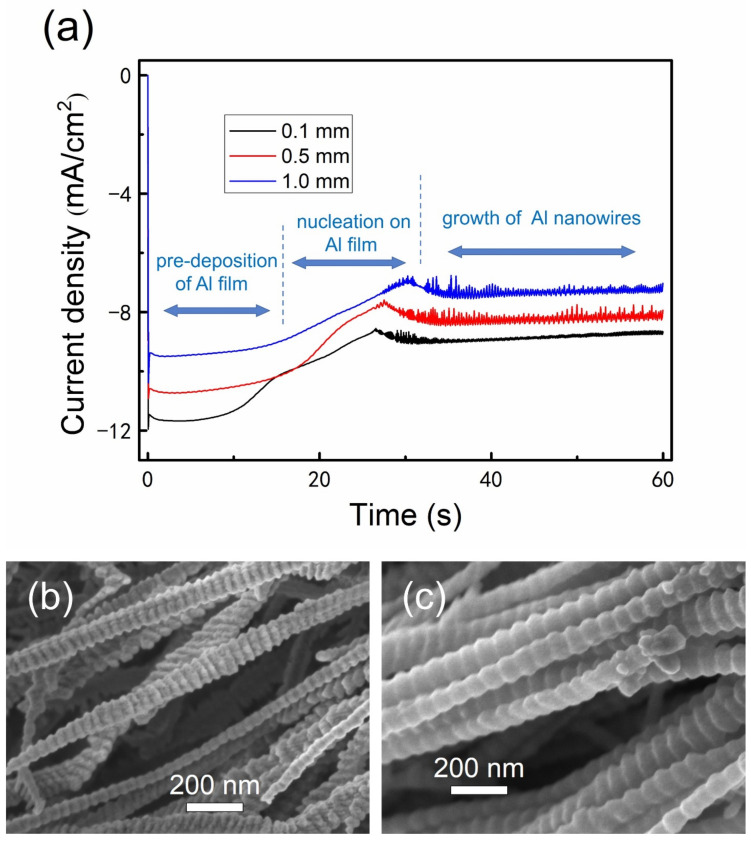
Potentiostatic electrodeposition of Al nanowires at −1.2 V/25 °C for 60 s under the restricted diffusion condition (inner diameter of the insulation ring was 8 mm). (**a**) Chronoamperometry curves recorded on Cu plate with different thicknesses of insulation ring. SEM images of the electrodeposited Al nanowires on the central part of Cu plate, using an insulation ring with a thickness of (**b**) 0.1 mm and (**c**) 0.5 mm.

**Figure 7 nanomaterials-12-01390-f007:**
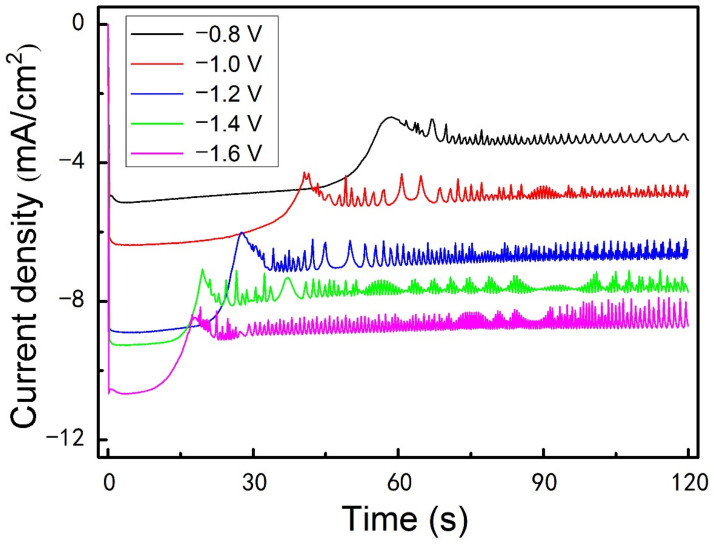
Chronoamperometry curves recorded on Cu plate during the electrolysis at various potentials under the restricted diffusion condition with a 2 mm–thick insulation ring (inner diameter was 6 mm) at 25 °C.

**Figure 8 nanomaterials-12-01390-f008:**
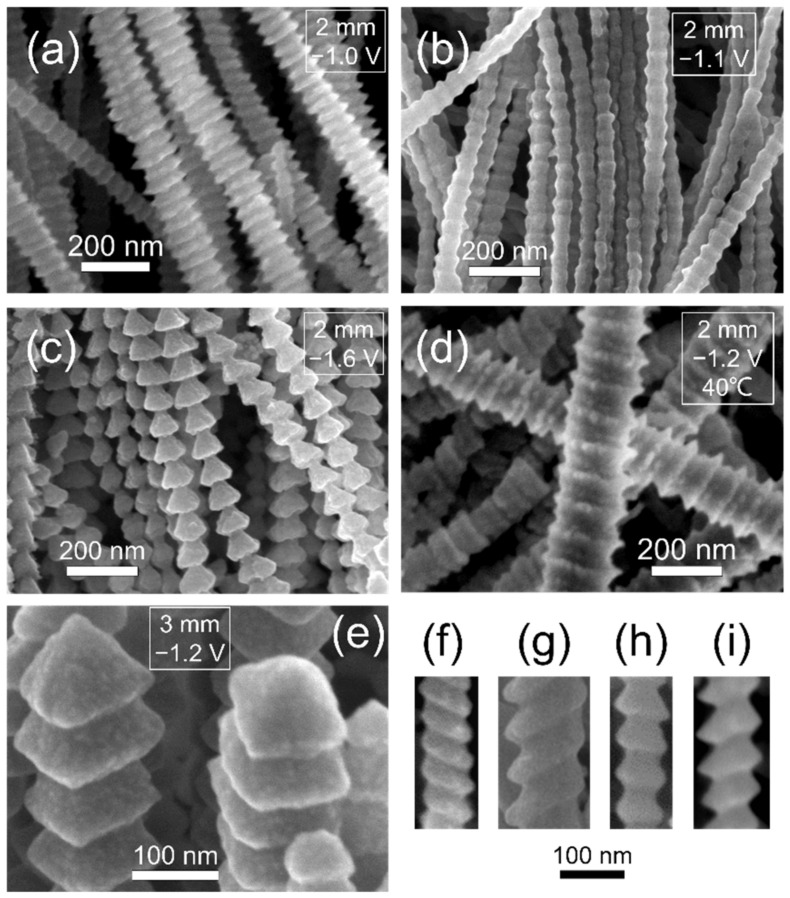
SEM images of Al nanowires electrodeposited at (**a**) −1.0 V/25 °C, (**b**) −1.1 V/25 °C, (**c**) −1.6 V/25 °C and (**d**) −1.2 V/40 °C under the restricted diffusion condition with a 2 mm–thick insulation ring; and (**e**) at −1.2 V/25 °C with a 3 mm–thick insulation ring. The inner diameter of the insulation ring was 6 mm. Some Al nanowires with other special morphologies: (**f**,**g**) spiral-like and (**h**,**i**) sloping spine-like.

**Figure 9 nanomaterials-12-01390-f009:**
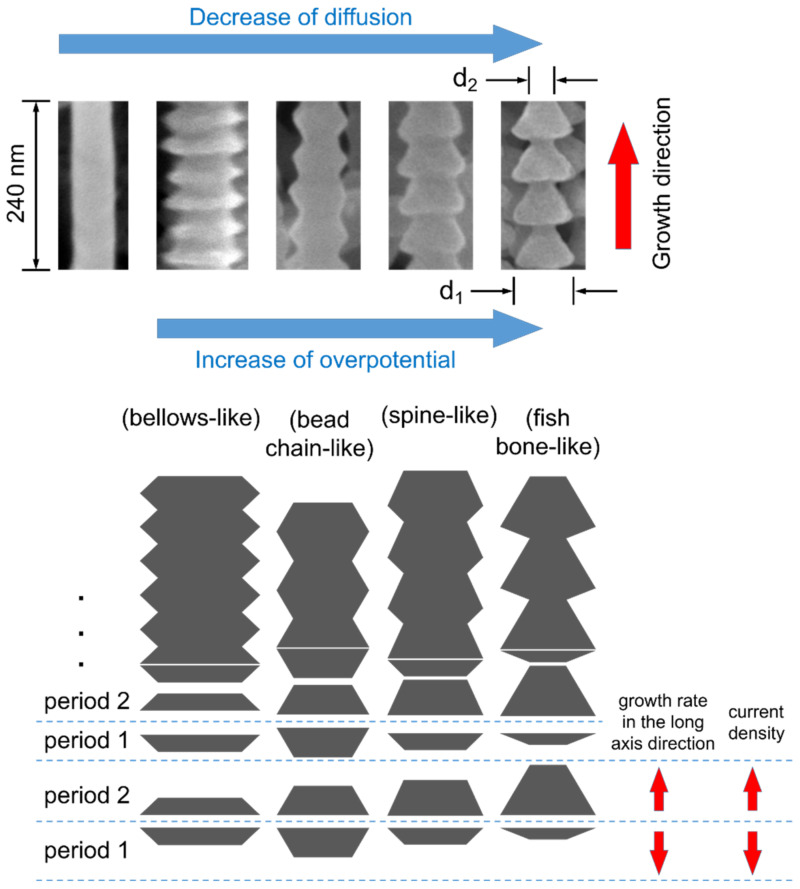
Typical morphologies of Al nanowires in SEM observation and the schematic of the growth process.

**Figure 10 nanomaterials-12-01390-f010:**
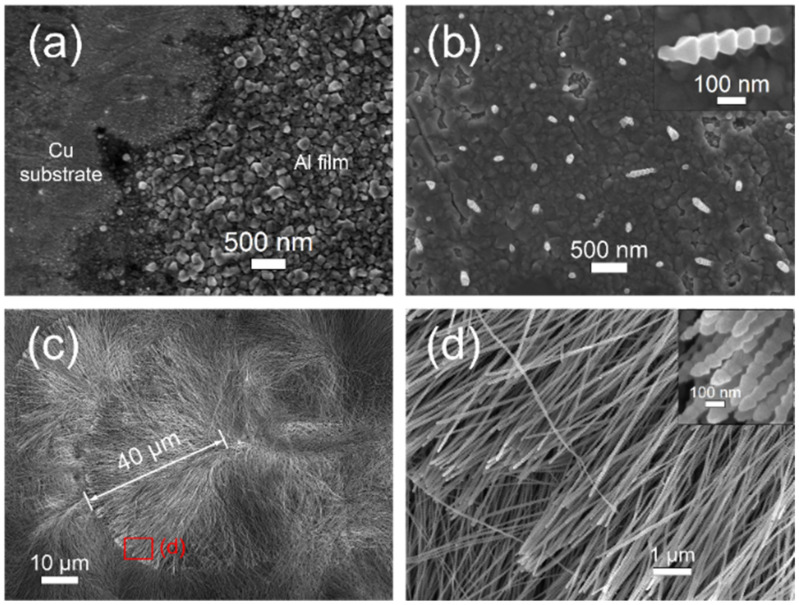
SEM images of the deposits obtained in potentiostatic electrodeposition at −1.4 V/25 °C (**c**,**d**) under the diffusion restricted condition with a 2 mm–thick insulation ring (inner diameter was 6 mm): (**a**) Al nano thin film on the Cu substrate after electrodeposition of 15 s, (**b**) Al nano-nuclei and short Al nanowires after electrodeposition of 25 s and (**c**,**d**) Al nanowires after electrodeposition of 300 s.

**Figure 11 nanomaterials-12-01390-f011:**
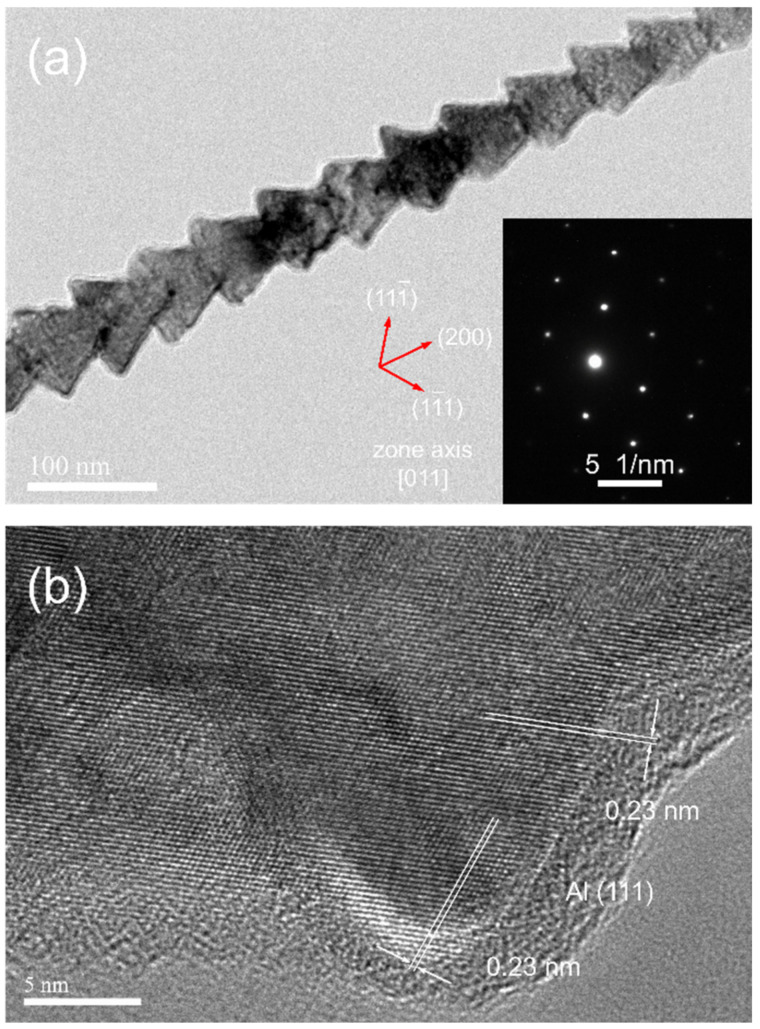
(**a**) TEM image, SAED pattern and (**b**) HRTEM image of a fish-bone-like Al nanowire electrodeposited at −1.6 V/25 °C for 120 s with a 2 mm–thick insulation ring (inner diameter was 6 mm).

**Table 1 nanomaterials-12-01390-t001:** Average current densities of plateaus and the decrease of diffusion flux, using different thicknesses of insulation ring (inner diameter was 6 mm) according to Figure 2a and Figure 4.

Thickness of Insulation Ring (mm)	0 (without Insulation Ring)	0.1	0.5	1	2	3
Average current density of plateaus (mA/cm^2^)	−21.43	−11.84	−10.80	−9.23	−6.79	−5.06
Decrease of diffusion flux (%)	0	44.75	49.6	56.93	68.32	76.39

## Data Availability

The data presented in this study are available upon request from the corresponding authors.

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
