# Peer review of "An Effective Strategy for Template-Free Electrodeposition of Aluminum Nanowires with Highly Controllable Irregular Morphologies"

_nanomaterials, 2022, doi:10.3390/nano12091390_

Round 1
Reviewer 1 Report
The manuscript by Wang et al. describes the possibility to alter the shape of electrodeposited aluminum nanowires by alteration of potential and deposition geometry. They find that, by using a PTFE ring surrounding the area of deposition, the diffusion of Al-species is limited and affects the shape of the wires. This limitation is also observed when increasing the potential. When limited by the diffusion of Al, the wires grow periodic lateral shapes, ranging from smooth, rough, beads and finally, in the authors’ words, fish bone-shaped. The study paves the way to grow nanowires with desired shapes and high-index facets, e.g. useful in catalytic applications. In general, the manuscript is nice and is graspable, making it attractive for a broad range of audience, and is well introduced with lots (!) of references to previous studies of growth, shape-control etc. of different materials.
The manuscript is a nice read, but there are some questions and comments below which I think should be answered or improved before being accepted:
General concepts:
- I find the last paragraph of the intro lacking in what was more specifically done here. Mention the physical constraints and potential adjustment specifically so the reader knows the scope of your study and what to expect in a clearer way. Additionally, perhaps create a subheading under “Materials and methods” in which you list the different measurements done. The results part is so long the reader needs a heads-up on what is coming.
- I am missing a concise conclusion on what parameters to use to get what shape, as a “take-home message”. I also lack some discussion/comment on the difference between the different parts of the hole. Can the geometry be changed to create a more homogenous distribution of wires (so they can be useful in applications)?
- I think the names you’ve chosen (screw, bead, vertebra etc) are a bit confusing and you should perhaps not mention them in the abstract (where the reader has not seen the images). Perhaps indicate which one is which in some form of schematic drawing (or typical SEM images, see later comment).
- The presentation of the results is quite long and sometimes difficult to grasp. I would suggest using subheadings like you did in the “Materials and methods”. g., “Effect of ring size”, “effect of overpotential”, “shape trends” etc or similar. Perhaps considering shorten it slightly
Specific comments:
- Figure 3 needs to be fixed: color of arrows (yellow on yellow) and resolution of their borders.
- Table 1: “Ratios” is not needed. Should be just “decrease” (also used in text and caption)
- Figures 4a and b: Why can’t these be combined for space saving? They are not overlapping. (if I’m not missing some reason for them to be split)
- You use the word “vertebra” which to my knowledge is only for one single bone in an animal’s spine. Wouldn’t it be more descriptive to use “spine” since we’re talking about a series of them (used in multiple places in the manuscript)?
- Figure 5: Difficult to grasp. Put some labels of the thicknesses (and voltages) used. Figure 8: also a bit unclear in terms of what trends we can see. Annotate what difference there is (overpotential) also in the figure.
- I am missing some schematic drawings of the mentioned periods (1 and 2) of growth (page 9), which I think are crucial in the explanation of the shape of the wire. And this should be accompanied with references in the text to the schematics so it’s easier for the reader to follow. This would also help the explanation in the first paragraph on page 11. Could drawings be included in figure 9 in some way? What shapes are we seeing and what is the progression (from low limitation to high limitation)?
- You mention the screw shape. That is difficult to see in the SEM-images. Does it really form a spiral? If so, why does a spiral form and why not always? How does it relate to the growth periods? Draw in figure 9 or in other potential schematics.
Reviewer 2 Report
The authors present a study on the evolution of Al nanowire morphology as a function of restricted diffusion conditions enabled by a restrictive PTFE ring. They investigate a range of restrictive ring dimensions and applied voltage and temperature parameters, and the morphology of Al nanowires evolve from smooth shape to fish-bone geometry. Authors propose a general method to controllably modify the irregular morphology of Al nanowires.
I would suggest the manuscript to go through a major revision process before being reconsidered for publication in Nanomaterials journal.
- Firstly, I would strongly recommend an edit of the overall manuscript. Some of the technical terms, chemical notations, scientific terminology and flow of the paragraphs can greatly benefit from a professional edit work.
- In Figure 3, the authors should descriptively demonstrate which geometrical parameters they alter (inner diameter, thickness etc.) by denoting with appropriate symbols/letters.
- In addition to using qualitative descriptions of bead-like, fishbone etc. the authors can benefit from defining quantitative morphological features of corrugation like diameter ratio of sections, periodicity and can present the correlation between electrodeposition parameters and these quantitative morphological parameters. (One suggestion is to use the parameters defined in Figure 9 throughout the text)
- The authors comment on the dynamics of the electrodeposition process using the evolution of chronoamperometry curves throughout the text. It would greatly benefit the manuscript, if authors can correlate the current density time evolution with nucleation, deposition etc. steps using a diagram in the supporting information.
- In Figure 4a, 0.1mm and 0.5mm curves present distinctive features, an additional shoulder, during current density ramp up. The increase is not as sharp as the curves of Figure 4b. It would be great if authors can elaborate on that.
- Figure 4 is also missing individual captions for a and b.
- Figure 5 is presenting a lot of data in a non-systematic manner. I find it difficult to distinguish among the different classes of varying parameters. For ex. why are there 4 images for 2mm, two different voltages and different deposition durations ? I would suggest authors to present the data in a more concise manner with additional notations i.e. arrow showing the direction of increasing ring thickness.
- The morphological change with increasing ring thickness in Figure 6 does not align with the dependency presented in Figure 5. I would appreciate if authors comment on that.
Round 2
Reviewer 1 Report
I think the authors have improved the manuscript and I am happy that it has been improved to this point where I think only minor questions/suggestions remain.
I especially like the nicely drawn schematics and references to these in the text for improved clarity.
Some minor remaining comments (will reference the numbering previously used), the rest looks good:
2. When I mentioned a conclusion and "take-home message" I meant also in the conclusion part (text). You say: "The decrease of diffusion leads to the appearance of particular segmented morphologies. As the diffusion decreases gradually, the morphology of Al nanowires turns from smooth surfaces to rough surfaces...". Can you just shortly mention how to achieve this (as you have described earlier) here as well?: What size and height of the ring will create what types of nanowires in your case?
7. I accept the explanation by the authors to separate the figures (fig 4) even if I still think it could have been done to save space, but ok as is now.
11. I still think it's a bit misleading to talk about a "screw" since there's no spiraling happening. A screw is for me mainly a spiraling object rather than the object you use in wood-work. It confused me as a reader, so I still suggest changing to something simpler as you have not created a spiraling object.
Author Response
- Thank you for the valuable advice. We have re-written the conclusion part according to the reviewer’s suggestion. All the changes in the manuscript have been marked in green.
- Thanks for the comments. We sincerely appreciate your understanding.
- Thanks a lot for your valuable suggestions. We have changed all the “screw” to “bellows” in the manuscript.
Reviewer 2 Report
I would like to thank the authors for the revisions. The changes significantly increased the data presentation quality and rendered the text easier to follow. I would recommend the manuscript to be accepted in the present form.
Author Response
Thanks a lot for the comments and valuable suggestions.